# Latent Space Representations for Marker-Less Realtime Hand–Eye Calibration

**DOI:** 10.3390/s24144662

**Published:** 2024-07-18

**Authors:** Juan Camilo Martínez-Franco, Ariel Rojas-Álvarez, Alejandra Tabares, David Álvarez-Martínez, César Augusto Marín-Moreno

**Affiliations:** 1Department of Industrial Engineering, Universidad de los Andes, Bogota 111711, Colombia; jc.martinez10@uniandes.edu.co (J.C.M.-F.); a.rojasa55@uniandes.edu.co (A.R.-Á.); a.tabaresp@uniandes.edu.co (A.T.); 2Integra S.A., Pereira 660003, Colombia; cmarin@integra.com.co

**Keywords:** computer vision, robotics, hand–eye calibration, deep learning, synthetic data, autoencoders

## Abstract

Marker-less hand–eye calibration permits the acquisition of an accurate transformation between an optical sensor and a robot in unstructured environments. Single monocular cameras, despite their low cost and modest computation requirements, present difficulties for this purpose due to their incomplete correspondence of projected coordinates. In this work, we introduce a hand–eye calibration procedure based on the rotation representations inferred by an augmented autoencoder neural network. Learning-based models that attempt to directly regress the spatial transform of objects such as the links of robotic manipulators perform poorly in the orientation domain, but this can be overcome through the analysis of the latent space vectors constructed in the autoencoding process. This technique is computationally inexpensive and can be run in real time in markedly varied lighting and occlusion conditions. To evaluate the procedure, we use a color-depth camera and perform a registration step between the predicted and the captured point clouds to measure translation and orientation errors and compare the results to a baseline based on traditional checkerboard markers.

## 1. Introduction

Reliable hand–eye calibration, performed to find the relationship between the frames of reference of a robot and a visual sensor or camera whether the latter is mounted on the end-effector (eye-in-hand) or statically with respect to the base of the robot (eye-to-hand), is often based on specialized markers or patterns with easily discernible visual features and known physical dimensions. Said relationship is typically described as a square transformation matrix where the coefficients of said matrix are estimated through the capture of several images and feature matching of the known markers in the robot and its workspace until a suitable projection model can be calculated or inferred. This process must be repeated when either the camera or the base of the manipulator is moved or rotated with respect to the other in eye-to-hand systems, which may prove cumbersome in highly dynamic workspaces [1].

In contrast, marker-less hand–eye calibration methods seek to find the calibration matrix relationship without the need for physical markers. This approach offers several advantages:Efficiency: With marker-less calibration, the robot can be recalibrated easily if the camera, the base of the robot (in eye-to-hand scenarios), or its end-effector is changed or repositioned, without the need to reapply physical markers.Flexibility: Marker-less calibration eliminates the need for specialized markers, reducing the cost of setup and maintenance as well as increasing the range of viable workspaces.Increased accuracy: Marker-less calibration techniques can sometimes offer higher accuracy compared to marker-based methods, especially in scenarios where markers may be difficult to detect.

Nevertheless, marker-less methodologies frequently employ depth data that may not be readily available. These approaches rely on computationally complex representations of the robot and the workspace, often obtained from specialized hardware or through expensive feature-matching algorithms. A more cost-effective process can be achieved through learning-based pipelines, but special care must be given when dealing with rotation predictions, as they are often difficult to regress directly.

## 2. Related Work

Marker-less calibration has seen significant interest in recent years. As depth sensors can digitally capture three-dimensional data that previously had to be regressed from 2D images become more available, this kind of data has become the cornerstone of several marker-less methods. In [2], stereo vision was used to match visual features that are used to estimate the centroids of geometries in the scene. In [3], several filtering procedures as well as iterative closest point steps were performed to match their rotation and translation prediction to point clouds captured with a structured light sensor, while in [4], a 3D scanner was used to obtain sub-millimeter-accurate scans to perform nearest-neighbor fitting of robot geometries.

Learning-based methods seek to offer less computationally and financially costly solutions [5]. In most cases, synthetic data have proven to be a powerful tool to support these approaches [6]. A frequently studied strategy consists of using visual features to detect keypoints in 2D space to relate with joint angles [7]. DREAM [8] predicted joint keypoint heat maps in the same manner as other authors performed human pose estimation, but they nevertheless relied on point cloud registration with depth data to correct their predictions. Robot joint configuration [9] and keypoint selection [10] for similar techniques frequently represent a challenge for symmetric robots.

## 3. Materials and Methods

Our work proposes a collection of methods that separately estimate the position and orientation of a robot with respect to a monocular RGB camera that is not attached to the said robot to construct a calibration matrix. The algorithms are run sequentially, beginning with the detection of the robot on the scene captured by the camera, followed by the prediction of the orientation parameters of the matrix, and finalizing with the prediction of the position of the robot that encompasses the final column of the matrix.

### 3.1. Object Detection

As the proposed calibration is designed to work without markers, it requires a method to find a region of interest (ROI) within the digital image that contains relevant visual information for the calibration procedure. While markers can often be used in pattern-matching algorithms, they are sensitive to occlusion, lighting, orientation, and scale variations [11]. Learning-based detectors, in contrast, have achieved state-of-the-art performance in most modern benchmarks. We use one such detector, YOLOv5 [12], that uses convolutional neural network architectures as a backbone, to detect the robot and, more granularly, the end effector within a 2D image. The detection can be visualized as a bounding box drawn on top of an RGB image, as shown in Figure 1.

The ROI serves two purposes: first, its location, size, and shape serve as parameters for a coarse position estimation algorithm based on a camera projection model; and second, a resized crop of the image in the shape of the ROI is used in the orientation estimation step, based on a different class of artificial neural network.

### 3.2. Orientation Estimation

The rotation part of the transform is typically difficult to estimate with sufficient precision using learning methods based on convolutional neural networks (CNNs) with linear outputs that treat orientation estimation as a classification or regression problem. It is speculated that certain views captured by the camera are prone to result in disproportionately larger errors due to the same visual features being shared across widely different rotations of the captured objects, particularly those with strong axial symmetry such as robotic manipulators. A possible option to train a learning-based model that focuses on structural features instead of discriminative ones is to use a fully convolutional architecture, as is the case of convolutional autoencoders.

### 3.3. Convolutional Autoencoders

Convolutional autoencoders (CAEs) and convolutional neural networks (CNNs) both use convolutional layers for feature extraction. However, CAEs are designed to learn a compressed representation (latent space) of input data, which can later be used to reconstruct the original input. The encoder part of the CAE learns to compress the input into a lower-dimensional representation, and the decoder part learns to reconstruct the original input from this representation (see Figure 2).

This process forces the model to capture the most important features of the input while discarding non-essential details. CAEs are often better at preserving structural information of a captured scene because they are explicitly trained to reconstruct the input. This means that the learned latent space representation is forced to encode the most salient features of the input, which often include structural information such as spatial relationships and, crucially, rotational transforms.

### 3.4. Latent Space Representation of Orientation

While the latent space representation used by the decoder block to reconstruct the original input likely contains the orientation information that is being sought, it also contains confounding information such as lighting and shading, data regarding background color and shape, visual noise, etc. A possible strategy consists in not training a traditional autoencoder, but rather a denoising autoencoder. Denoising autoencoders do not attempt to reconstruct the original image, but rather a version of that image stripped of some sort of visual feature, typically noise.

By using an autoencoder that reconstructs an image containing only visual cues regarding the orientation of the object, as shown in Figure 3, we prioritize the representation of orientations in the latent space. This is the method used by Sundermeyer et al. to perform 6DoF object detection of rigid objects, naming this architecture augmented autoencoder (AAE) [13]. The decoder portion of the AAE is only used during training, so the encoder will utilize considerably fewer computational resources during inference compared to the training phase.

### 3.5. Orientation Codebook

Regression of the latent space representation z suffers from the same pitfalls as CNN architectures. However, it has been shown that similar AAE representations reconstruct similar orientations [14]. Therefore, given a lookup table of *k* known $z_{i}$ representations paired with the known rotation parameters $\alpha_{i}$, $\beta_{i}$, and $\gamma_{i}$ they represent, it is possible to find the closest $z_{i}$ to a measured representation $z_{m}$, where $\alpha_{i}$, $\beta_{i}$, and $\gamma_{i}$ approximately equal $\alpha_{m}$, $\beta_{m}$, and $\gamma_{m}$. The difference between $z_{\mathrm{ref}}$ and $z_{m}$ is described by the cosine distance ${\mathrm{dist}}_{\cos}$ (Equation (1)) between them.
(1)$${\mathrm{dist}}_{\cos}\left(A,B\right)=1-\frac{A\cdot B}{\left|A\right||B|}$$

This discretized lookup table, shown in Table 1, which we call the orientation codebook, cannot fully represent the continuous orientation space, but if it is constructed with sufficient granularity, we believe we can achieve sufficiently small orientation errors in the calibration procedure.

### 3.6. Camera Projection Models

Three-dimensional coordinates may be represented in a 2D space using different projection models. While orthographic projections will always display an object with the same projected area regardless of distance from the object to the camera, in perspective projections the area and general shape of the captured objects will vary as the distance from the camera changes. These variations are described by the pinhole camera projection model (visualized in Figure 4) and the intrinsic parameters of the camera.

The focal length *f* and sensor size variables *W* and *H* govern the field of view of the camera, with shorter focal lengths and larger sensor sizes resulting in a higher field of view, where a greater portion of 3D geometry may be projected to a 2D image without changing its size. These parameters are fixed and can be used to determine the *u* and *v* coordinates in the image plane of a given point in three-dimensional *x*, *y*, and *z* coordinates (see Equation (2)).
(2)$$u=f\frac{x}{z}\;v=f\frac{y}{z}$$

Given a known physical ∆x or ∆y between two points where the value of *z* is unknown but remains approximately constant in both, the said value can be solved for when a corresponding ∆u or ∆v is available, due to triangle similarity. Conversely, if *z* is known along with ∆u and ∆v, the unknown ∆x and ∆y may be found. In fact, the individual coordinate values for bounding box points may be found this way, as described by Equations (3) and (4).
(3)$$\Delta u=u_{2}-u_{1}=f\frac{x_{2}}{z}-f\frac{x_{1}}{z}=f\frac{\Delta x}{z}$$
(4)$$z=f\frac{\Delta x}{\Delta u}=f\frac{\Delta y}{\Delta v}$$

In the case of a robot arm (where CAD geometry is usually available) paired with a camera with known intrinsic parameters, it is possible to establish an approximate relationship between the size of the projected bounding box of the robot on the captured image and the coordinates of the *x* and *y* edges of the robot for a reference pose (orientation and translation) relative to the camera. The projection for such a pose or view is displayed in Figure 5.

For the given bounding box on the camera view with a resolution of 1920 × 1080 px, the $z_{\mathrm{ref}}$ distance is 2.5 m, ${∆u}_{\mathrm{ref}}$ is 264 px, and ${∆v}_{\mathrm{ref}}$ is 724 px (see Figure 6). Model data indicate that, for this particular pose, ${∆x}_{\mathrm{ref}}$ is 0.527 m and ${∆y}_{\mathrm{ref}}$ is 1.391 m. From Equation (4), as long as the orientation portion of the reference pose remains relatively unaltered, any change in translation that results in a new $z_{\mathrm{new}}$ will have ${∆u}_{\mathrm{new}}$ and ${∆v}_{\mathrm{new}}$ follow the relationship:(5)$$z_{\mathrm{new}}/z_{\mathrm{ref}}={∆u}_{\mathrm{new}}/{∆u}_{\mathrm{ref}}={∆v}_{\mathrm{new}}/{∆v}_{\mathrm{ref}}$$

### 3.7. Position Codebook

By following the same procedure used to find the size relationship for one view of the robot, it is possible to establish the same relationship for multiple views of it. By saving a sufficiently large set of camera poses that can plausibly be found in the workspace to a dictionary-like structure, along with their corresponding bounding box ∆u and ∆v, an initial estimate for the *x*, *y*, and *z* coordinates of the virtual box corners may be found for a new detection bounding box.

However, as hand–eye calibration is performed in relation to a coordinate frame placed on the robot, offset parameters ∆h and ∆w should be added to the dictionary, relating the positional transform between the corners or center of the bounding box and the target origin (see Figure 7).

These variables share the same relationship of similarity exhibited by the size of the bounding box. Visual representations of the records included in such a dictionary are displayed in Table 2.

We name this dictionary the position codebook.

### 3.8. Real-Time Calibration

The calibration procedure, as described by Algorithm 1, is performed in three stages: detection (lines 1–3), orientation estimation (lines 4–11), and position estimation (lines 12–14). First, a bounding box is obtained from the object detector (line 2), assuming there is a sufficiently complete view of the manipulator identified with sufficient confidence. This region is cropped and, if necessary, padded with black pixels to achieve a square input, which is then fed to the encoder (line 4). The encoder produces a latent space vector, which is matched to the closest value in the orientation codebook, compared by cosine similarity (lines 7–11), as suggested by [13].

The corresponding rotation is used as a key to retrieve the ${∆u}_{\mathrm{ref}}$ and ${∆v}_{\mathrm{ref}}$ from the position codebook, where the projection parameters are estimated (line 13) through $z_{\mathrm{new}}$, found by substituting ${∆u}_{\mathrm{new}}$ and ${∆v}_{\mathrm{new}}$ in Equation (5) with the values obtained from the object detector bounding box along with the established $z_{\mathrm{ref}}=$1 m. By Equation (4), with $z=z_{\mathrm{new}}$, *x* and *y* are estimated after adding the ∆h and ∆w values from the position codebook. Finally, the estimated rotation and translation transforms are combined into the final hand–eye calibration transform (line 14). Algorithm 1 can be run on a loop, with a standby or continue condition given by the presence of a detection of sufficient confidence (line 3).
**Algorithm 1.** Continuous Hand–Eye Calibration
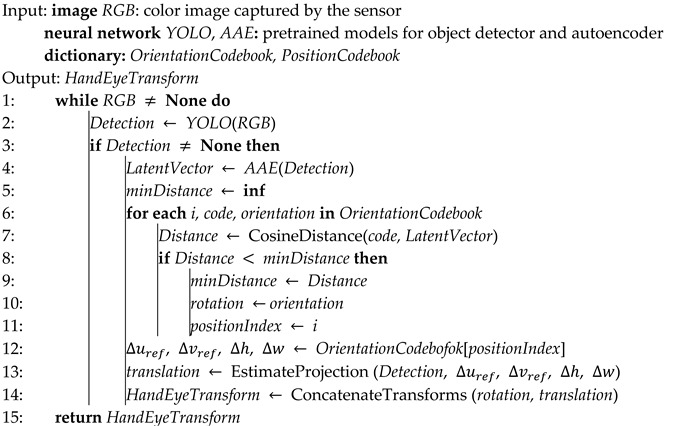


The position and orientation coordinates are combined into the matrix *T* (robot to camera) of the robot with respect to the coordinate frame of the camera, which can be inverted to transform from camera coordinates to robot coordinates. The proposed calibration procedure may prove useful in path-planning algorithms designed to accommodate unstructured or highly variable environments [15].

### 3.9. Dataset

All computer vision algorithms benefit from, if not require, precisely annotated data in the three-dimensional domain. Such data can be difficult or prohibitively expensive to obtain, which is why we opted for completely synthetic data to train the different models (shown in Figure 8) except for the object detection model, which uses a mix of real and synthetic samples.

Synthetic data were created using available CAD models converted to meshes for use in Blender [16], an open-source 3D modeling and animation tool enhanced with automated computer vision annotation scripts [17]. We believe that the robust physically based rendering (PBR) capabilities of Blender serve to bridge the reality gap [18] that degrades the performance of models trained with generated data. Nevertheless, we also implemented domain randomization techniques, both during the setup of the virtual scenes and as post-processing effects. This enhances the capability of the models to generalize, as real-world inputs are interpreted as an additional domain variation among the ones used to train the models [19].

## 4. Computational Experiments and Results Analysis

### 4.1. Model Training

The YOLO detector was trained on a dataset of 1100 images, 1000 synthetic and 100 manually annotated, over 100 epochs. Images were scaled and padded to the 640 × 640 px size expected as inputs by the detector. Using an 80/20 train and validation split, the model achieved perfect recall and accuracy, albeit this could stem from uniformity in the distribution of lighting conditions and spatial configurations seen in the dataset.

The autoencoder was trained for 200 iterations on 10,368 synthetic samples, 8640 produced by domain randomization to be used as inputs and 1728 serving as labels to be reconstructed. The training was performed using the Adam optimizer and mean squared error (MSE) loss. Learning rate decay from 0.0001 to 10^−6^ was implemented to prevent overshooting and overfitting, and early stopping conditions were defined to improve the capability of the model for generalization, but overall performance saw little change towards the end of training, even as the validation loss kept decreasing up to the last epoch. The training progression for some of the orientations considered during training is visualized in Figure 9. Both models were trained using the CUDA API on an NVidia GTX 1060 GPU with 6 GB of video memory.

### 4.2. Experimental Calibration Setup

Calibration will be performed on a Universal Robots UR10 cobot, a serial manipulator with six degrees of freedom. As we aim to perform monocular calibration and tracking and to eliminate the necessity for markers, both the proposed approach and the classical marker-based techniques used to obtain a baseline are calculated using the monocular RGB projection of a digital camera. Ground truth annotations are easily attainable for simulated data, also constructed in Blender, while real-world experiments are performed using a Kinect V2 RGB-D camera. Standard RGB camera calibration procedures were followed to obtain camera intrinsics and correct for radial distortion through the methods available on the OpenCV library. The focal length of the Kinect V2 forces a distance of at least 2 m between the sensor and the robot to ensure the latter fits within the images captured by the sensors. The obtained color information is fed to the proposed and baseline models, while the depth information is used as the target geometry while performing a registration procedure based on the iterative closest point (ICP) algorithm. The latter calculates a transformation matrix that describes the experimental error, in millimeters for the position deltas and with the rotation matrix converted to Euler angles.

Flexibility to changes in the workspace, the main area we expect to improve upon from marker-based methods, is tested by displacing the sensor to six different positions with respect to the robotic manipulator. On each new position, both the baseline and the novel method produce estimations for the robot pose with respect to the camera frame. Both virtual and real-world setups are visualized in Figure 10. As the estimates are obtained, we measure the computation time as well as the absolute position and orientation errors. Both the marker-based method and the proposed approach produce deterministic results on simulated data, while physical runs were repeated 10 times for each camera position.

### 4.3. Baseline Using Checkerboard Markers

To identify any improvements, advantages, and disadvantages of the proposed calibration procedure, we established a hand–eye calibration baseline obtained from traditional marker-based methods [20,21]. In eye-to-hand scenarios like the one being studied, a pattern is attached to the end effector of the robot and moved to n different poses to obtain a set of end-effector transforms ${H_{\mathrm{eff}}^{\mathrm{Base}}}_{i}$ where 0<i<n with respect to the robot base. For each pose, the camera must be able to capture the pattern along with visual features to calculate the transforms ${H_{\mathrm{pattern}}^{\mathrm{camera}}}_{i}$ where 0<i<n with respect to the camera frame. This is a camera calibration problem, where known geometry and different views are leveraged to overcome the loss of dimensionality that occurs during image projection [22]. A constant but unknown transform (unless the exact geometry of the coupling to the end effector is known) $H_{\mathrm{pattern}}^{\mathrm{eff}}$ completes the spatial relationship required to know $H_{\mathrm{camera}}^{\mathrm{base}}$ and may be solved for with the Tsai-Lenz method [23]. This algorithm is frequently referenced when evaluating against marker-based calibration methods [24,25] and has been found to be particularly robust to translational noise within that category [26].

The different views are obtained by setting the robot to 12 different configurations that are always used regardless of camera position.

### 4.4. Accuracy

The measured error values for the simulated and real experiments, and for the classic and the proposed marker-less procedures, are given in Table 3 and Table 4. In camera positions where eight or more views of the checkerboard markers were available, errors were significantly lower compared to the marker-less prediction. However, in positions where the checkerboard is detected on fewer views, the error value rises dramatically. This is probably caused by the partial acquisition of features, which is known to degrade the performance of Tsai’s algorithm [26]. The marker-less methodology maintains similar error rates across different camera poses. Performance translated reasonably well from simulated to real-world data, suggesting successful bridging of the reality gap, at least for this application.

The calibration accuracy resulting from the proposed method is generally lower than the one reported on similar robots by marker-less methods such as DREAM [8]. This, however, only holds true for the matrices obtained prior to the iterative closest point step. The approach presented in this work achieves a 100% registration success rate on a few iterations of the ICP algorithm, greatly improving calibration accuracy when depth data are available.

### 4.5. Flexibility

The proposed method displays substantially higher resilience to the occlusion of segments of the robot body. To alleviate the difficulties brought upon by self-occlusion in the marker-based scenario, bespoke joint configurations would likely be required to keep the markers in the view of the camera while being distinct enough to get data to optimize for the different transforms. Although the checkerboard markers are never removed from the robot throughout the course of the experiments, they would need to be made use of the end effector, only to be mounted again to recalibrate the robot–camera relationship. This could result in a laborious process compared to temporarily modifying the joint pose of the robot once, the sole requirement to follow the marker-less procedure. Additionally, compared to other CNN-based methods found in the literature, the AAE used in our approach has approximately 8 million trainable parameters, in contrast to architectures relying on VGG19 which have at least 144 million parameters [8,27]. The lower computational requirements allow the presented model to be deployed on a broader range of hardware.

All detection, pose estimation, and calibration procedures were executed on a Windows PC with an Intel Core i5 12400 CPU with 16 GB of RAM and an Nvidia GTX 1060 GPU.

### 4.6. Perspective Distortion

Depending on the physical characteristics of a digital camera (the intrinsic parameters) and its position and orientation in space with respect to the captured scene or object (the extrinsic parameters), the images obtained may show different size relationships between the projected objects. This phenomenon is known as perspective distortion and potentially affects the performance of the learning models for position and orientation estimation. Consider the two images shown in Figure 11. The top view shows that the robots share the same rotation transform with respect to the camera, as well as the same *z* distance to the camera plane, but have different *x* and *y* components on their respective translations.

Due to the change in perspective, the projections are considerably different, as shown in Figure 12. These projections are encoded into different latent space vectors and result in different orientation predictions, which implies one or both views are susceptive to perspective distortion. Moreover, any error during orientation prediction can lead to the wrong key being used to retrieve projection data from the position codebook, resulting in additional errors being added to the position prediction.

Careful study of this phenomenon must be conducted to reduce this source of error, but we found that a way to ameliorate its effects is to maintain the projection of the robot close to the center of the camera view.

## 5. Conclusions

We proposed an ensemble of methods to perform hand–eye calibration between a robot arm and a camera mounted on the world, an eye-to-hand scenario. This approach exploits salient visual features, known three-dimensional geometry, and projected information to predict the pose and orientation of the robot with respect to the camera from monocular RGB images without the need for fiducial markers.

The proposed methods were tested both on simulated data and a real-world workspace with an RGB-D sensor and were found to be resistant to occlusions and position and orientation changes of the camera. Additionally, the components of the ensemble can process new inputs in real time. This leads to increased flexibility and adaptability to dynamic workspaces when compared to traditional techniques that rely on physical markers. However, a combination of ambiguous features, a highly discretized prediction space, and susceptibility to perspective distortions harm the accuracy of our approach.

These obstacles may be addressed by increasing the granularity of possible predictions, standardizing the capture procedure, and by using depth data to refine the predictions. Additionally, as the detector model can identify and crop multiple instances of the robot within the scene, it is possible that hand–eye calibration can be performed for multiple robots simultaneously. Further comparisons with state-of-the-art marker-less methods can help identify other strengths and weaknesses of our approach.

## Figures and Tables

**Figure 1 sensors-24-04662-f001:**
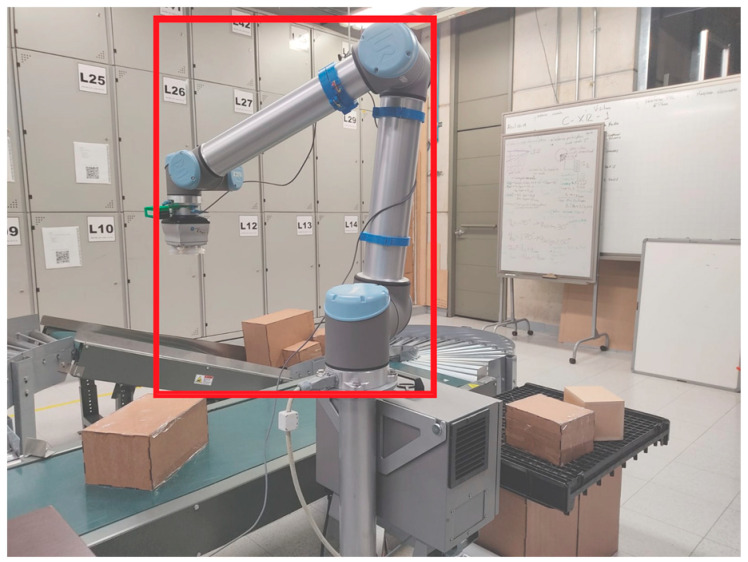
The bounding box in red encloses only the geometry strictly belonging to the robot.

**Figure 2 sensors-24-04662-f002:**
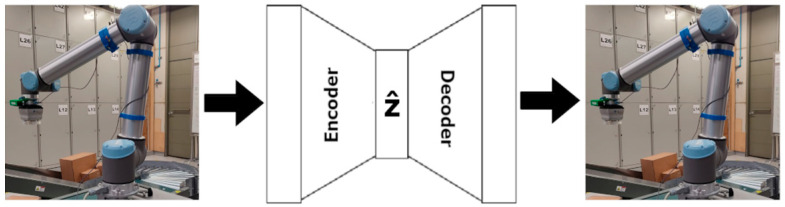
Convolutional autoencoders reduce the dimensionality of an input (an image in this case) to the size of a latent vector *ẑ* on the encoder and then reconstruct the original input with the decoder.

**Figure 3 sensors-24-04662-f003:**
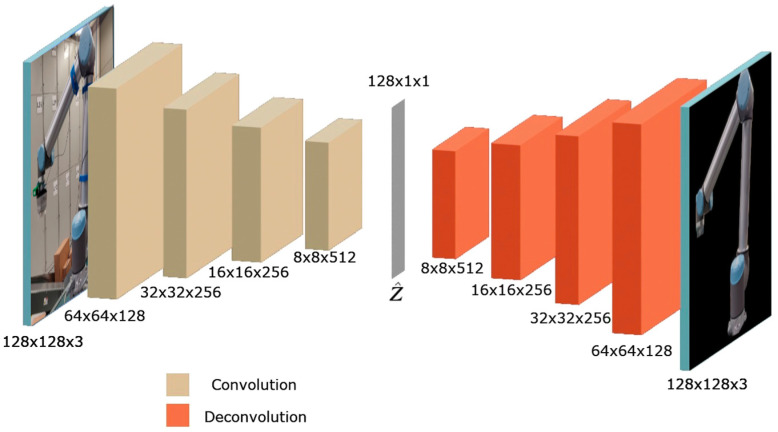
The AAE architecture used to construct the latent vector. The parameters for convolution and deconvolution operations are based on [13].

**Figure 4 sensors-24-04662-f004:**
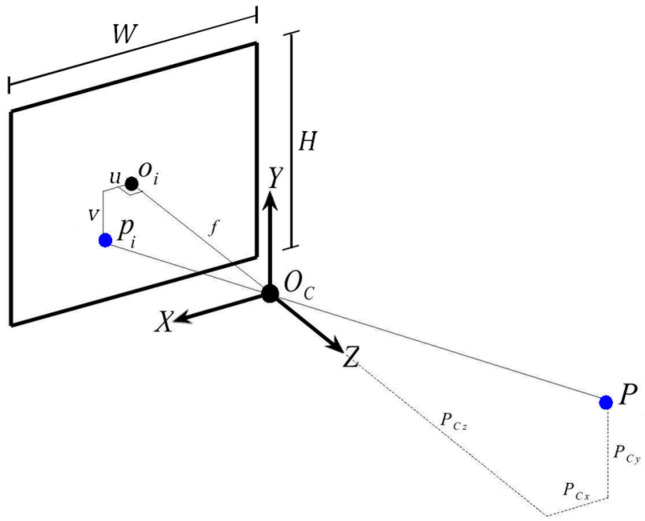
The virtual image plane is visualized in front of the camera center.

**Figure 5 sensors-24-04662-f005:**
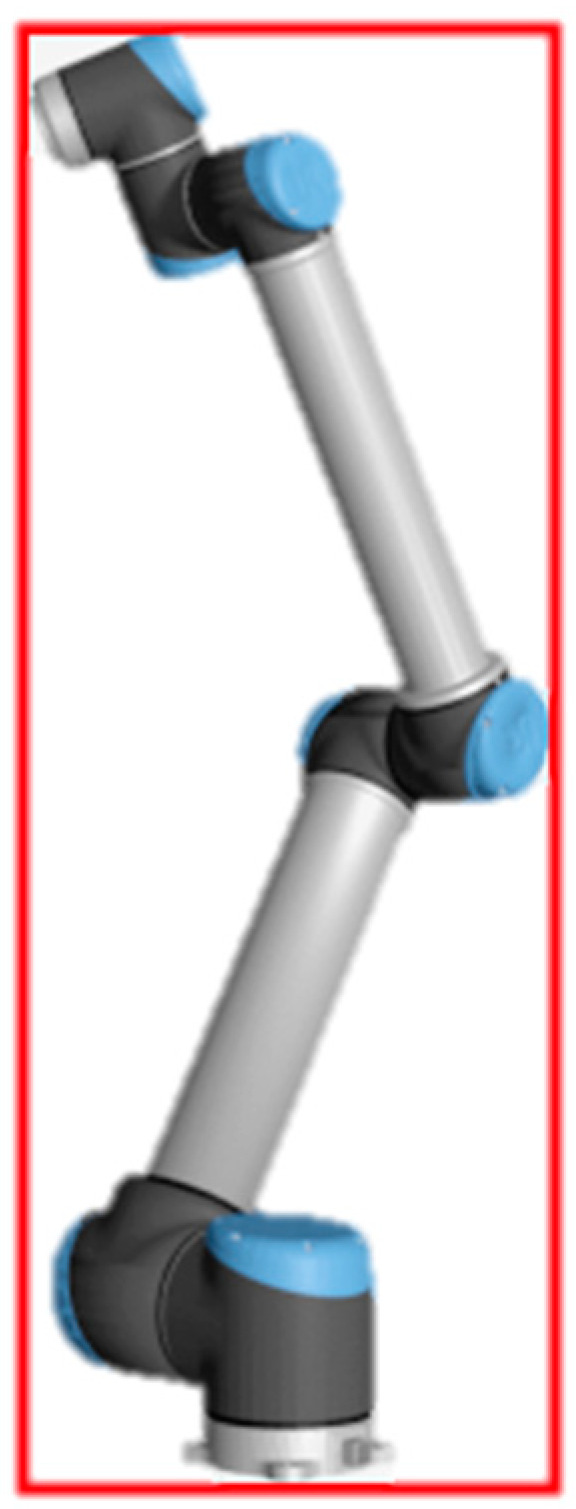
A reference projection of the robot arm. ∆u and ∆v are the horizontal and vertical sizes of the bounding box in camera coordinates.

**Figure 6 sensors-24-04662-f006:**
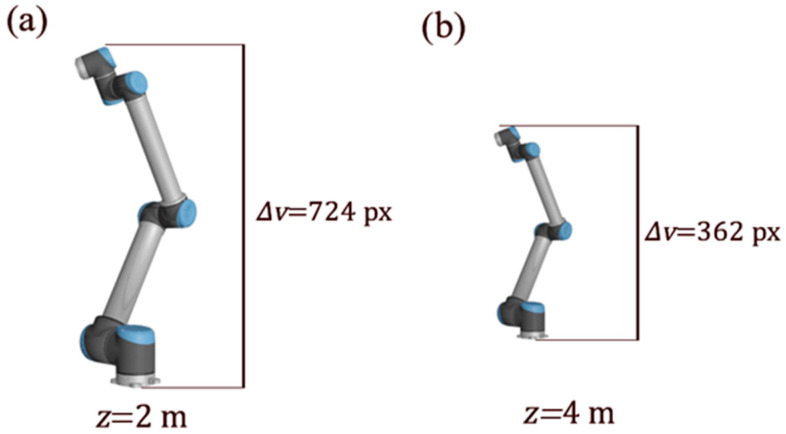
(**a**) Projection height in pixels for a distance of 2 m to the camera plane and (**b**) projection height at 4 m. Notice how, at half the distance, the projection size is twice as tall.

**Figure 7 sensors-24-04662-f007:**
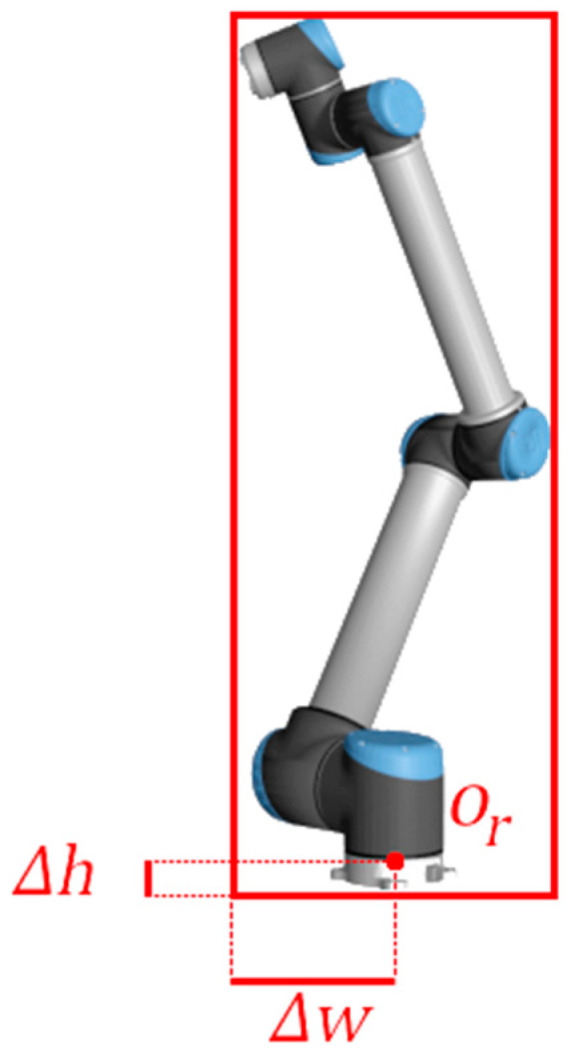
The projected origin of the base of the robot is not aligned with the center of the bounding boxes.

**Figure 8 sensors-24-04662-f008:**
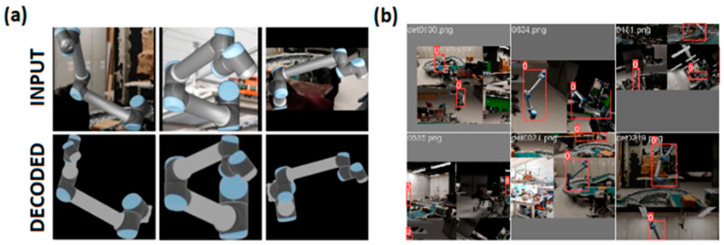
(**a**) Synthetic data points for the autoencoder and (**b**) object detection models.

**Figure 9 sensors-24-04662-f009:**
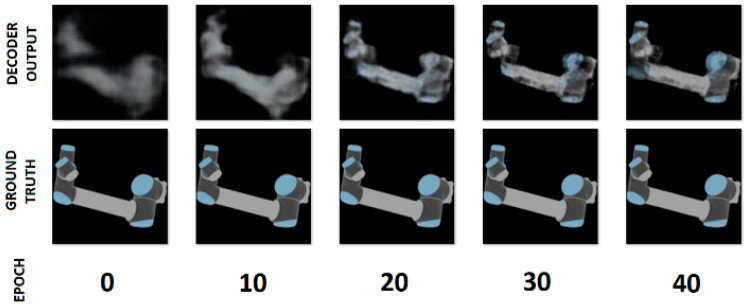
Reconstruction progress for the AAE.

**Figure 10 sensors-24-04662-f010:**
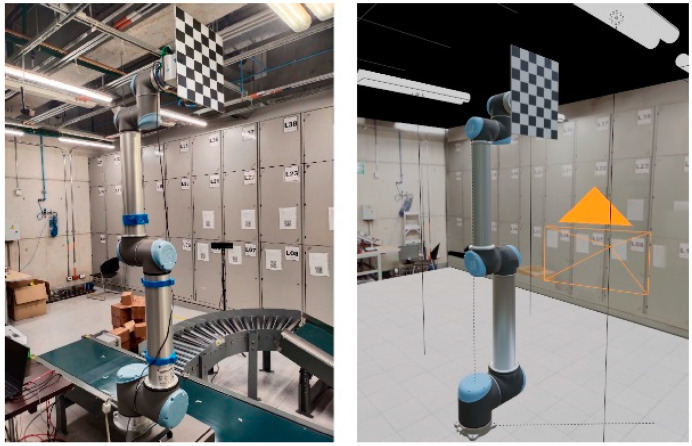
Experimental setup, both real and simulated within Blender.

**Figure 11 sensors-24-04662-f011:**
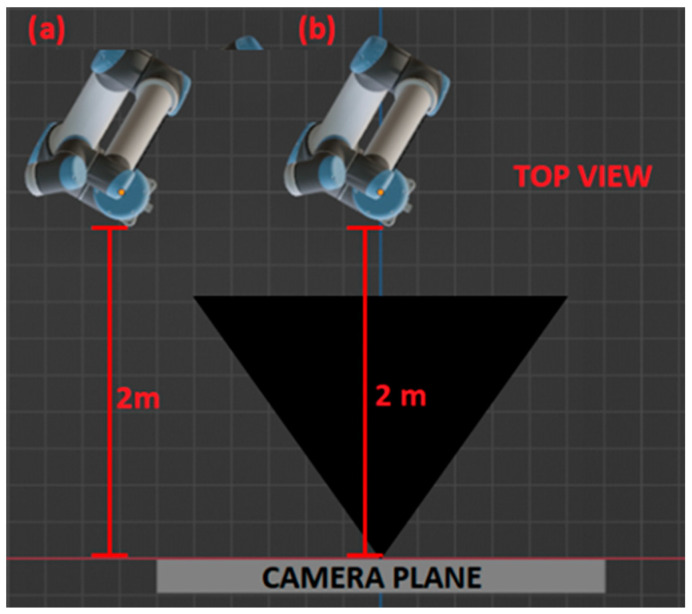
The robot maintains the same rotation with respect to the camera coordinate frame, only the *x* coordinate is modified, *z* remains constant.

**Figure 12 sensors-24-04662-f012:**
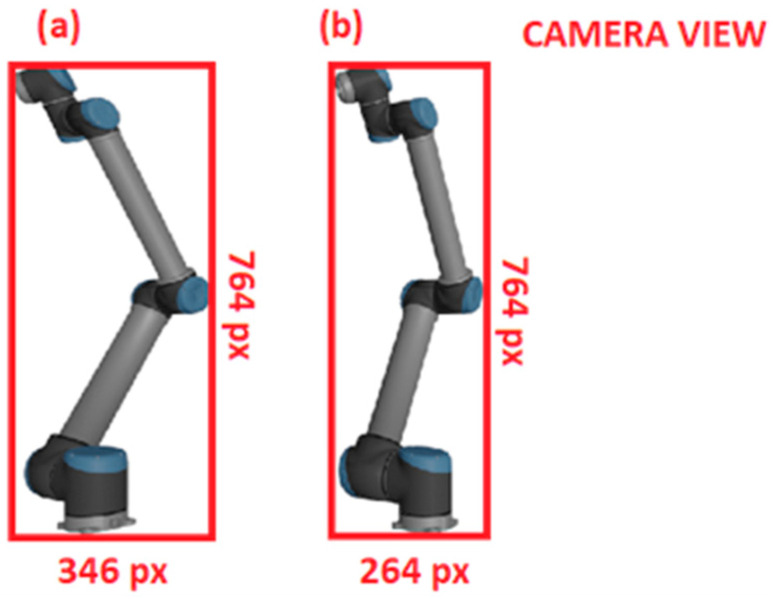
The bounding boxes have different aspect ratios and are encoded into different latent vectors, even though both views share the same rotation transform.

**Table 1 sensors-24-04662-t001:** An example of a lookup table that associates the latent space vector $z_{i}$ to a set of rotation parameters (Euler angles).

i	$$z\in R^{L}$$	α[rad]	β[rad]	γ[rad]
1	$z_{0}$	0	0	0
2	$z_{1}$	0.1745	0	0
…	…	…	…	…
k−1	$z_{k-1}$	π	π	3.054
k	$z_{N_{or}}$	π	π	π

**Table 2 sensors-24-04662-t002:** Sample records of the position codebook.

i	∆x	∆y	∆h	∆w
1	224	725	10	20
2	229	725	10	21
…	…	…	…	…
*K* − 1	701	430	15	700
*k*	711	430	15	705

**Table 3 sensors-24-04662-t003:** Calibration results for real-world data.

Camera Position	Translational Error (mm)		Rotation Error (Degrees)		Computation Time (ms)		Calibration Time (s)	
	Tsai	Ours	Tsai	Ours	Tsai	Ours	Tsai	Ours
1	26.95	**26.95**	**0.0318**	14.7390	425.29	**6.25**	18.500	**1.624**
2	**13.53**	34.89	**0.0199**	14.0361	395.87	**6.17**	18.496	**1.714**
3	239.36	**48.75**	**4.8322**	17.4627	251.96	**6.26**	18.358	**1.682**
4	362.69	**37.84**	**8.6518**	20.1085	248.01	**5.67**	18.332	**1.662**
5	191.82	**21.49**	**0.5433**	22.7529	263.20	**6.20**	18.367	**1.540**
6	123.81	**20.86**	**2.1312**	19.7310	298.50	**6.14**	18.521	**1.516**

**Table 4 sensors-24-04662-t004:** Calibration results for simulated data.

Camera Position	Translational Error (mm)		Rotation Error (Degrees)		Computation Time (ms)		Calibration Time (s)	
	Tsai	Ours	Tsai	Ours	Tsai	Ours	Tsai	Ours
1	**13.30**	18.25	**0.0239**	5.3456	417.07	**6.04**	0.449	**0.006**
2	**9.14**	23.61	**0.0144**	5.0092	403.65	**6.28**	0.419	**0.007**
3	136.48	**27.82**	**2.6591**	3.6061	256.16	**6.31**	0.275	**0.007**
4	178.93	**19.74**	**2.6459**	5.2111	252.10	**6.05**	0.272	**0.007**
5	111.82	**11.83**	3.3087	**3.1251**	268.72	**6.12**	0.285	**0.007**
6	72.24	**12.65**	**1.2087**	4.8164	303.88	**6.29**	0.322	**0.007**

## Data Availability

The original data presented in the study are openly available in UR10-Detection at https://github.com/jcmartinez10/UR10-Detection (last accessed on 5 June 2024).

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
