# Peer review of "Latent Space Representations for Marker-Less Realtime Hand–Eye Calibration"

_sensors, 2024, doi:10.3390/s24144662_

Round 1

Reviewer 1 Report

Comments and Suggestions for Authors

The article deals with topical problem and brings new findings and inspire approach to solving the problem of hand-eye calibration of robotic manipulator using monocular RGB camera.

The structure of the article and the level of presentation are very well.

I have no serious objections.

I would like just to ask the authors to add following information:

1. Add the information what is optimal combination of the camera system parameters (detector resolution, lens focal length, the object distance) with respect to the size of the calibrated robot to achieve optimal (accurate) results of calibration.

2.  What version of YOLO detector has been used in this work?

3. What calibration method has been used for the determination of the RGB camera intrinsic parameters? Was the lens distortion taken into account?

4. Can this method be used for hand-eye calibration of multiple robots by one camera simultaneously?

Reviewer 2 Report

Comments and Suggestions for Authors

1. The introduction of an augmented autoencoder neural network for marker-less hand-eye calibration in unstructured environments represents a significant advancement in robotics research. The approach holds promise due to its computational efficiency and ability to operate in varied lighting and occlusion conditions.

2. While the concept of leveraging latent-space representations for improved orientation estimation is intriguing, additional details on the network architecture and training process are essential for a comprehensive understanding. Providing insights into the design choices and hyperparameter selection would enhance the reproducibility and applicability of the proposed technique.

3. The evaluation methodology, focusing on registration between predicted and captured point clouds, offers valuable insights into translation and orientation errors. However, to strengthen the paper's impact you can read and cite [1] A survey of interval observers design methods and implemen- tation for uncertain systems”. Journal of the Franklin Institute, 358(6), pp. 3077-3126. [2] Set-Membership Interval State Estimator Design using Observability Matrix for Discrete-Time Switched Linear Systems”. IEEE Sensors Journal, 20(11), pp. 6121-6129.

To further validation through experiments in diverse real-world scenarios is recommended. This could involve environments with varying levels of complexity and occlusion, as well as comparisons against existing state-of-the-art methods beyond traditional marker-based calibration approaches.
